# Mapping qualitative research on motor imagery: A scoping review

**Juliet M. Rowe**[1,2], **Theresa C. Gaughan**[1,2], **Shaun G. Boe**[1,2,3]*

1 Laboratory for Brain Recovery and Function, Dalhousie University, Halifax, Nova Scotia, Canada, 2 PhD Health Program, Faculty of Health, Dalhousie University, Halifax, Nova Scotia, Canada, 3 School of Physical Therapy, Western University, London, Ontario, Canada

* sboe@uwo.ca

## Abstract

### Aim

To examine the extent to which qualitative methods have been used in motor imagery research, and to characterize how these approaches have been applied.

### Design

Scoping review conducted in accordance with JBI methodology.

### Methods

Eligible articles included peer-reviewed literature investigating motor imagery using qualitative methods. Screening and data extraction were conducted in Covidence. Data were synthesized descriptively and presented in tabular form.

### Results

Thirty-nine articles met the inclusion criteria. Most were situated in sport psychology ($n=23$; 59%), with athletes comprising the most frequently studied population ($n=20$; 51.3%). Over half of the qualitative studies ($n=20$; 51.3%) did not report a specific research design. The predominant focus was on how specific populations use motor imagery ($n=27$; 69.2%), while only one study (2.6%) investigated learning through motor imagery.

### Conclusion and Impact

The literature is heavily weighted toward sport psychology and understanding athletes' use of imagery, with limited attention to how motor imagery is experienced in experimental settings. Notably, few studies have examined the process of imagery – particularly its role in learning – within laboratory contexts. Given that such laboratory research often forms the foundation for the application of motor imagery in healthcare, including rehabilitation, the absence of qualitative insights into participants'

**Data availability statement:** All data are in the manuscript and supporting information files. Final extracted data and tabulated results are openly available in the Open Science Framework repository (https://osf.io/4rcbd/?view_only=fec-de6e44f794e8a8c6c08e2c7046d19).

**Funding:** This work was supported by an NSERC Discovery Grant (RGPIN/04840-2020) awarded to SB and personnel support from NSERC (Canada Post Graduate Scholarship – Doctoral) and the Heart and Stroke foundation (BrightRed Student Research Award) awarded to JR. The funders were not involved in any aspects of the work or its dissemination. The funders had no role in study design, data collection and analysis, decision to publish, or preparation of the manuscript.

**Competing interests:** The authors have declared that no competing interests exist.

experiences represents an important gap. Clearer reporting of qualitative designs and more in-depth exploration of motor imagery experiences are needed to ensure that applications, particularly in rehabilitation, are grounded in the perspectives of those engaging in motor imagery practice.

## Introduction

Motor imagery, the imagination of movement in the absence of physical execution, is an effective tool for skill acquisition [1–4]. This makes it a promising strategy for rehabilitation following brain injuries that impair motor function, such as stroke. However, evidence for its efficacy in recovery following stroke is mixed [5,6]; a recent meta-analysis reported only moderate efficacy [7]. One reason for this inconsistency may be that the processes underlying motor imagery, particularly motor imagery-based learning, remain poorly understood.

Several theories propose mechanisms of learning through motor imagery (for a review see Hurst & Boe [8]). Early frameworks suggested that motor imagery functions as a simulation of overt movement, recruiting the same brain regions and processes, thereby supporting skill acquisition [1]. Yet direct evidence for this broad theory is limited. To test such theories, researchers rely primarily on behavioural and neurophysiological measures: behavioural measures of motor imagery-based learning include changes in reaction time in implicit sequence learning tasks [9], measures of accuracy in kinematically complex drawing tasks [3], and presence of aftereffects in prism adaptation paradigms [10,11]; neurophysiological measures, including transcranial magnetic stimulation [12,13], functional magnetic resonance imaging [14,15], and magneto/electroencephalography [16–18], have advanced understanding of the neural correlates of motor imagery. Collectively, these approaches have refined and diversified theoretical accounts over the last two decades.

Despite these advances, a central challenge persists: motor imagery is an inherently covert process. It is accessible only to the individual performing it and cannot be directly observed [19]. Understanding how people subjectively experience and engage with motor imagery is therefore crucial. Yet most of the existing literature is quantitative, leaving experiential dimensions underexplored. One potential contributor to this gap lies in the dominant paradigmatic assumptions shaping the field – the foundational beliefs about the nature of reality, how knowledge is measured, and what knowledge is valued [20]. Motor imagery research has largely developed within post-positivism, privileging hypothetico-deductive designs aimed at identifying generalizable truths about phenomena [20]. While such approaches have produced important advances, they also limit what is considered knowable, measurable, and valued within the field. In turn, this constrains the types of questions asked, influences funding priorities, and shapes the availability of qualitative research training. Consequently, researchers often rely on assumptions about motor imagery experiences – for example, that instructions to "imagine what it would look like and feel like to…" effectively engage both visual and kinesthetic modalities. However, participants may experience motor imagery differently, perhaps drawing on other forms

of imagery (e.g., auditory) and/or perspectives (first-person versus third-person) that behavioural and neurophysiological measures cannot capture [21,22].

To address the challenge of studying a covert process, motor imagery research has employed methods consistent with post-positivism. These include indirect behavioural measures such as mental chronometry (comparing imagined and executed movement times scaled to task demands) [23–25], movement times across experimental groups [10,11], and self-reported accuracy ratings [26]. In addition, tools such as the Kinesthetic and Visual Imagery Questionnaire (KVIQ) [27] and the revised Motor Imagery Questionnaire (MIQ-R) [28] were developed to assess imagery ability across experimental groups. Although valuable, these metrics cannot fully capture participants' subjective experiences: What exactly are they seeing? Feeling? Which modalities and perspectives are they using? Given the dominance of quantitative research within the field, the participant experience of this inherently covert phenomenon remains unknown. Further, because motor imagery functions as the primary experimental variable in imagery-based learning studies, unexamined variation in how imagery is experienced directly influences the conclusions about its efficacy and underlying mechanisms of learning. Consequently, translation to applied contexts such as rehabilitation and sport rests on researchers' assumptions about imagery processes instead of on participants' experiences of motor imagery. This disconnect has the potential to influence design, uptake, and overall effectiveness of motor imagery in applied contexts.

Qualitative research offers a valuable approach for addressing this gap by capturing the rich, experiential dimensions of motor imagery [29]. Specifically, qualitative inquiry seeks to understand and explore human experience through the collection and analysis of non-numerical data, with researchers constructing meaning from the data and generating themes or patterns [30]. Despite the widespread reliance on motor imagery in both sport and rehabilitation, the extent to which qualitative research methods have been used to investigate motor imagery is unknown. A scoping review is well-suited to broad research questions, mapping the extent, range, and nature of existing evidence, identifying gaps, and informing future directions [31]. Applying this approach therefore allows us to synthesize how qualitative approaches have been applied to motor imagery research to date and highlight opportunities for further inquiry.

## Methods

### Aim

The aim of this scoping review was to examine the extent to which qualitative methods have been used in motor imagery research, and to characterize how these approaches have been applied. The primary research question guiding the review was: *To what extent has motor imagery been investigated using qualitative approaches?* Secondary questions included: *In what settings, research fields, and populations has motor imagery been investigated using qualitative approaches? What types of qualitative methodologies have been used to investigate motor imagery? And what aspects of motor imagery are being studied (i.e., experiences, training, effectiveness of intervention)?*

### Design

This review was conducted in accordance with the JBI methodology for scoping reviews [31]. Findings were reported following the Preferred Reporting Items for Systematic Reviews and Meta-Analyses extension for Scoping Reviews (PRISMA-ScR) guidelines [32]. An a priori protocol was developed and preregistered on the Open Science Framework [33]. In accordance with recommended practices for scoping reviews, this study synthesized evidence from previously published sources and did not involve human participants. Ethics approval was therefore not required.

Inclusion criteria was developed using the Participant, Concept, Context (PCC) framework, as outlined in the review protocol [33]. Briefly, all human participants were eligible, including children, adults, individuals from clinical populations, and disabled persons; no exclusion criteria were applied based on participant characteristics. The central concept was motor imagery, defined as the mental rehearsal of movement without overt execution [1]. Studies focused on action

observation or combined action observation and motor imagery were excluded. Further, studies involving imagery that was not explicitly motor (e.g., visual-only, auditory, etc.) were excluded. The context encompassed studies using qualitative approaches to investigate motor imagery. Studies employing exclusively quantitative methodologies were excluded. Eligible sources included peer-reviewed research articles, as well as theses and dissertations. Only English language literature was included, and no date restrictions were set for the evidence sources.

### Search methods

A comprehensive search strategy was developed in consultation with a health sciences librarian to capture key concepts related to motor imagery and qualitative research (S1 File). Search strings were adapted to meet the indexing and syntax requirements of the following databases: MEDLINE [Ovid], PsycINFO [EBSCOhost], CINAHL [EBSCOhost], SPORTDiscus [EBSCOhost], Embase [Elsevier], and Dissertations and Theses Global [ProQuest]. All databases were searched on May 15th, 2025. To supplement the database search, a citation screening was conducted where the reference lists of all included studies were manually reviewed to identify additional eligible articles not captured in the initial search.

Search results from the databases were exported as .RIS files and uploaded into Covidence (Veritas Health Innovation, Melbourne, Australia) for screening. Additional articles identified through manual reference list review were first uploaded to Paperpile (web version 83; Paperpile LLC, Cambridge, MA), then exported as .RIS files and imported into Covidence for screening. Duplicates were automatically and manually removed in Covidence. All reviewers pilot screened the same 10 titles and abstracts independently, followed by a discussion to resolve discrepancies and clarify doubts and uncertainties. Once consensus was reached, title and abstract screening proceeded with each source independently reviewed by two reviewers (JR, TG). Subsequently, five full-text articles were pilot screened by all reviewers, reaching the same conclusions on eligibility of these articles. Full-text screening was then conducted independently by two reviewers (JR, TG), with predefined exclusion criteria documented within Covidence (S2 File). Conflicts that arose in both the title and abstract and full-text screening stages were resolved through discussion. When consensus could not be reached, a third reviewer (SB) was invited to make a final decision.

### Data extraction

A data extraction tool was developed in Covidence and piloted by all reviewers on three studies to ensure consistency and accuracy. The tool was refined iteratively during this process (S3 File). Two reviewers (JR, TG) independently extracted data from all included studies. In cases requiring consensus, JR manually verified the extracted data. Most discrepancies were minor, involving formatting or spacing; when substantive differences arose, JR and TG met to resolve through discussion. Extracted data included study characteristics, primary aims, methodological approaches, data collection methods, and participant details.

### Data analysis

Study characteristic data were analyzed using descriptive statistics. Findings were synthesized and organized into a table and graph for clarity and comparison.

Final extracted data and tabulated results are openly available in the Open Science Framework repository (https://osf.io/4rcbd/?view_only=fecde6e44f794e8a8c6c08e2c7046d19).

### Results

A search of six databases yielded 3129 articles that were uploaded to Covidence for title and abstract screening. After duplicate removal, 2098 articles remained. Following initial screening, 102 full-text articles were retrieved and screened independently by two reviewers, resulting in 36 articles that met the eligibility criteria. An additional 26 articles were

identified through citation screening of the included studies; four were retrieved for full-text review, and three met the inclusion criteria. In total, 39 articles were included in the final review (Fig 1).

## Study characteristics

Table 1 provides a summary of the study characteristics for the 39 included articles. The majority were primary research articles (*n* = 29; 74.4%), with the dominant methodological approach being qualitative (*n* = 31; 79.5%) and the remainder employing mixed methods (*n* = 8; 20.5%). Over half of the included studies did not specify an explicit qualitative research design (*n* = 20; 51.3%), however some studies drew on phenomenology (*n* = 5; 12.8%), grounded theory (*n* = 3; 7.7%), case study (*n* = 4; 10.3%), and a motor imagery-specific framework: four Ws of imagery (*n* = 2; 5.1%). Interviews were the primary method of data collection (*n* = 24; 61.5%) and geographically, most studies were conducted in Europe (*n* = 20; 51.3%) and North America (*n* = 18; 46. 2%).

## Disciplinary field, context, and population

Sport psychology represented the dominant disciplinary field investigating motor imagery through qualitative methods (*n* = 23; 59.0%) (Table 1) (Fig 2). Health-related disciplines accounted for approximately one-third of the studies (*n* = 13; 33.3), encompassing medicine, physiotherapy, kinesiology, and health sciences. Less dominant research fields included education (*n* = 1; 2.6%) and music (*n* = 2; 5.1%).

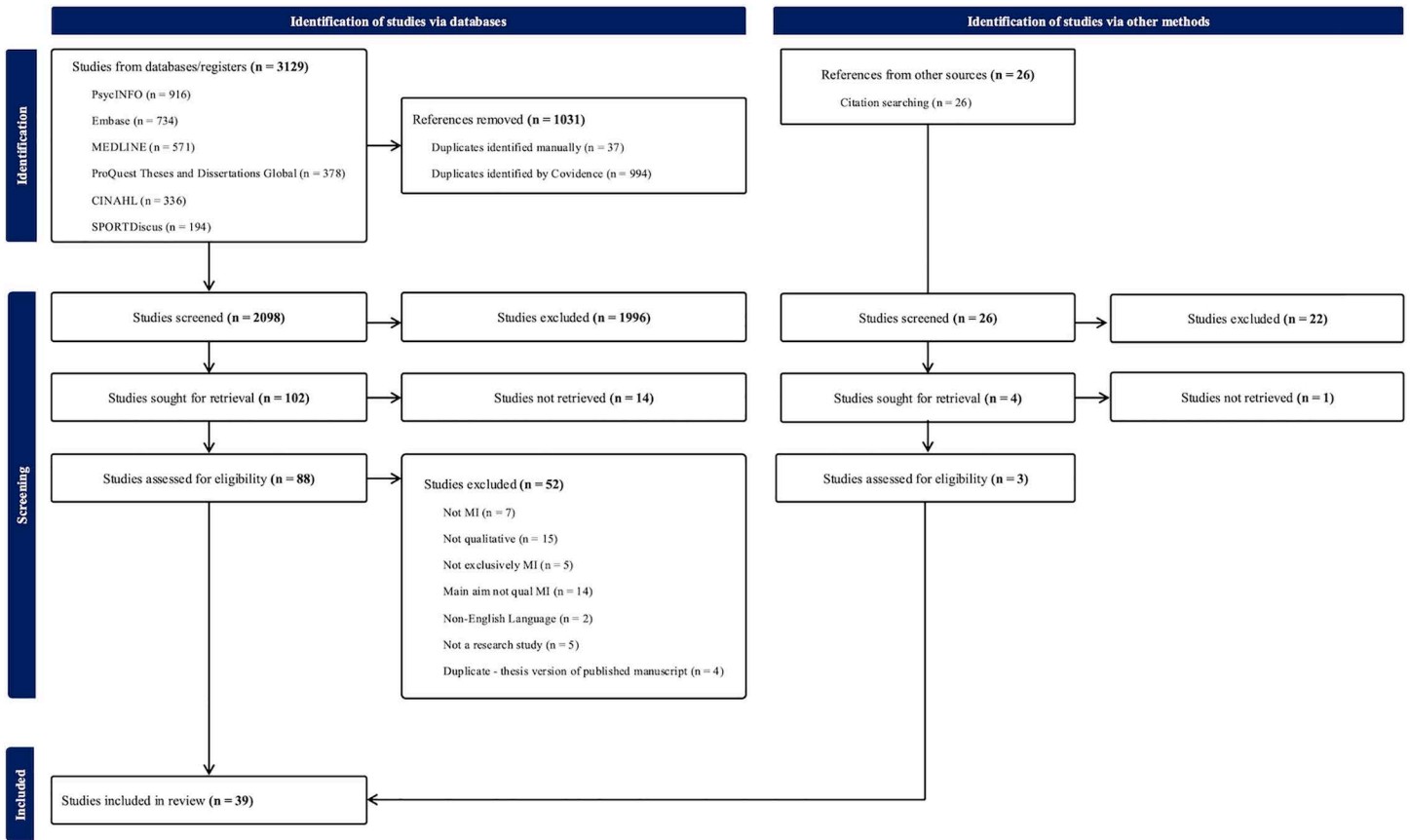

**Fig 1. PRISMA flow diagram.**

**Table 1. Overview of study characteristics of included studies.**

| Study Characteristics | n / 39 (%) |
|---|---|
| Type of Evidence Source | |
| Primary research article | 29 (74.4) |
| Evidence synthesis | 3 (7.7) |
| Thesis/dissertation | 7 (18.0) |
| Methodology | |
| Qualitative | 31 (79.5) |
| Mixed methods | 8 (20.5) |
| Design | |
| Unclear/Not specified | 20 (51.3) |
| Phenomenology | 5 (12.8) |
| Thematic analysis | 1 (2.6) |
| Four Ws of imagery framework | 2 (5.1) |
| Grounded theory | 3 (7.7) |
| Case study | 4 (10.3) |
| Exploratory | 1 (2.6) |
| Not applicable (not primary research) | 3 (7.7) |
| Method of Data Collection | |
| Interviews | 24 (61.5) |
| Multiple Methods | 9 (23.1) |
| Focus Groups | 2 (5.1) |
| Observation | 1 (2.6) |
| Not applicable (not primary research) | 3 (7.7) |
| Region where studies were conducted | |
| Europe | 20 (51.3) |
| North America | 18 (46.2) |
| Oceania | 1 (2.6) |
| Population | |
| Athletes | 20 (51.3) |
| Patients | 4 (10.3) |
| Musicians | 2 (5.1) |
| Dancers | 5 (12.8) |
| Clinicians | 5 (12.8) |
| Pilots | 1 (2.6) |
| Adults | 2 (5.1) |
| Context | |
| Sport and coaching | 16 (41.0) |
| Rehabilitation | 9 (23.1) |
| Recreational exercise | 2 (5.1) |
| Dance | 4 (10.3) |
| Education | 8 (20.5) |
| *Surgical* | *4 (50.0)* |
| *Aviation* | *1 (12.5)* |
| *Music* | *2 (25.0)* |
| *Medical* | *1 (12.5)* |
| Disciplinary Field | |
| Sport Psychology | 23 (59.0) |

*(Continued)*

**Table 1.** (Continued)

| Study Characteristics | n / 39 (%) |
|---|---|
| Health | 13 (33.3) |
| *Medicine* | *5 (38.5)* |
| *Physiotherapy* | *2 (15.4)* |
| *Kinesiology* | *2 (15.4)* |
| *Health Sciences* | *4 (30.8)* |
| Education | 1 (2.6) |
| Music | 2 (5.1) |
| Aspects of Motor Imagery Investigated | |
| Use of motor imagery | 27 (69.2) |
| *Athletes'* | *19 (70.4)* |
| *Exercises'* | *2 (7.4)* |
| *Dancers'* | *3 (11.1)* |
| *Surgical training* | *1 (3.7)* |
| *Music teachers'* | *1 (3.7)* |
| *Medical Faculty* | *1 (3.7)* |
| Experiences of motor imagery-based intervention | 8 (20.5) |
| Learning through motor imagery | 1 (2.6) |
| Development of imagery ability | 1 (2.6) |
| Role of imagery in dance | 1 (2.6) |
| Kinesthetic imagery processes | 1 (2.6) |

Italicized subcategories represent proportions of their corresponding main category.

Consistent with sport psychology's prominence, the most common research contexts were sport and coaching ($n=16$; 41.0%), followed by rehabilitation ($n=9$; 23.0%). Additional contexts included recreational exercise ($n=2$; 5.1%), dance ($n=4$; 10.3%), and education settings ($n=8$; 20.5%), the latter spanning diverse domains such as surgery, aviation, music, and medicine.

Study populations were generally highly specific (S4 File). The exception was two studies (5.1%) that examined motor imagery among an adult sample that engaged in exercise [34,35]. Athletes constituted the largest participant group ($n=20$; 51.3%), with other populations including musicians ($n=2$; 5.1%), dancers ($n=5$; 12.8%), clinicians ($n=5$; 12.8%), patients ($n=4$; 10.3%), and pilots ($n=1$; 2.6%).

## Aspect of motor imagery investigated

The most frequently examined aspect of motor imagery in qualitative research was its use within specific populations ($n=27$; 69.2%) (Table 1), of which the majority focused on athletes ($n=19$; 70.4%). Participants' experiences of motor imagery-based interventions represented the second most common focus ($n=8$; 20.5%). Notably, only one of the 39 included studies directly investigated learning through motor imagery [36], indicating that this aspect remains largely unexplored in the qualitative literature. Other less frequently examined topics included the development of motor imagery ability ($n=1$; 2.6%), the role of motor imagery in dance ($n=1$; 2.6%), and kinesthetic imagery processes ($n=1$; 2.6%).

## Discussion

This scoping review mapped and characterized how qualitative methods have been applied to the study of motor imagery. Across the 39 included articles (29 primary research articles, three evidence syntheses, and seven theses), the vast

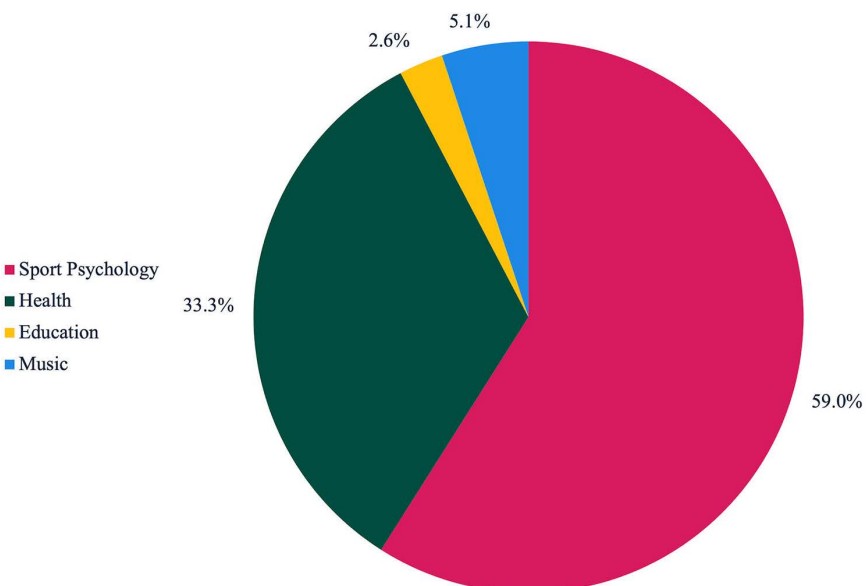

**Fig 2. Disciplinary distribution of included studies.**

majority were situated in sport psychology, where researchers primarily used interviews to better understand athletes' use of imagery. These studies largely focused on highly specialized groups, such as elite sky divers [37], expert golfers [38], elite canoe-slalom athletes [39], and female collegiate gymnasts [40], among others. Such work demonstrates how qualitative approaches can capture the experiences of athletes for whom imagery is integral to performance and training. Though sport applications represent the largest proportion of qualitative studies in motor imagery research, this review also highlights a diverse, but less developed body of research exploring motor imagery in other domains, including but not limited to rehabilitation [41–44], surgical training [45,46], aviation education [47], and music education [48,49]. The breadth of these applications highlights the versatility of motor imagery and demonstrates the need to further investigate how people in diverse contexts experience and make sense of this covert process.

A key gap identified in this review is the near absence of qualitative research investigating the experience of motor imagery, particularly its role in motor learning. Of all included studies, only one qualitatively examined learning through imagery – focusing on orthopedic trauma surgery training with the aim of developing imagery protocols for surgeons [36]. While valuable, this work remains applied rather than fundamental. To date, no laboratory-based motor learning research using behavioural or neurophysiological methods has integrated qualitative approaches to examine how individuals experience imagery during learning. This gap is critical because quantitative, post-positivist designs rest on assumptions about what participants' experience when instructed to perform imagery. For instance, how can researchers be certain that participants are practicing motor imagery as instructed, such as engaging only kinesthetic or visual modalities? Studies frequently distinguish between visual and kinesthetic motor imagery, reporting, for example, that kinesthetic-only imagery modulates corticomotor excitability [50], that tennis players preferentially use visual imagery over kinesthetic imagery [51], or that kinesthetic imagery aligns more closely with motor execution while visual imagery resembles motor observation [52]. However, without qualitative insight, it remains unclear whether participants genuinely restrict their imagery to the instructed modality, or whether they simultaneously draw from multiple forms of imagery. By overlooking participants' subjective experiences, research risks conceptualizing imagery as a uniform, controllable process, rather than acknowledging its inherently covert and subjective nature.

A second key finding of this scoping review was that more than half of the included articles did not report using a specific qualitative research design. While qualitative inquiry is often more flexible than quantitative approaches, it nonetheless relies on systematic methodological processes to ensure rigor. Selecting and articulating a qualitative design, such as phenomenology, grounded theory, or ethnography, is essential as these frameworks shape the formulation of research questions, strategies for data collection, analytic procedures, and standards for evaluation [30]. Without a clearly stated design, it becomes difficult to assess core indicators of rigor. For instance, when applying the *JBI Critical Appraisal Checklist for Critical and Interpretive Research* [53], it is impossible to determine whether a study's philosophical perspective aligns with its methodology if neither is reported. The same issue arises when assessing congruity between the research question, methodology, and analytic approach. Ultimately, this review emphasizes the need for future qualitative studies of motor imagery to explicitly report methodological theories and frameworks, ensuring transparency in paradigm, positioning, and design. In addition, future work would benefit from the systematic appraisal of existing qualitative literature to more thoroughly examine issues of methodological quality and rigor across the field.

Overall, qualitative methods in motor imagery research have been embraced within sport psychology, while remaining underexplored in other important contexts such as rehabilitation, education, and learning. Within rehabilitation, in particular, there has been a broader surge of qualitative inquiry, demonstrating its importance in the continued growth in the field [54]. In this scoping review, rehabilitation was the second most common context to employ qualitative methods. If qualitative methods are essential for capturing the experiences of patients using motor imagery in rehabilitation, they should also be integrated into basic laboratory research as the effectiveness of motor imagery-based rehabilitation ultimately depends on a grounded understanding of motor imagery-based learning processes. An in-depth exploration of motor imagery experiences is needed to ensure that application of motor imagery, especially in rehabilitation, are grounded not only in the quantitative outcomes, but also in the perspectives of those who engage in this inherently subjective and covert process. An exploratory sequential mixed methods design is one example of how researchers can integrate participant experience into experimental design. This approach is suitable for contexts where a phenomenon is not yet well understood and existing quantitative instruments do not adequately capture the phenomenon [55]. In this design, researchers first use qualitative methods to explore how individuals experience motor imagery. Insights from this phase informs the refinement or development of motor imagery measures grounded in participants' experiences. These refined tools are then evaluated in a large-scale quantitative motor imagery-based learning study. This approach will advance the field through grounding motor imagery research tools in the experiences of those who practice it.

While this scoping review addresses the need for qualitative inquiry in motor imagery research, the structural barriers to embracing pluralistic forms of knowledge production – particularly given the dominance of post-positivism – do not go unnoticed. Addressing these gaps requires effort across the research ecosystem: (1) institutions must provide training in qualitative theory and methodology; (2) journals must create space for qualitative inquiry within motor imagery and related disciplines; and (3) funding agencies must support research that investigates the experiential dimensions of inherently covert phenomena, such as motor imagery.

## Limitations

Although this review followed the JBI guidelines for scoping reviews [31] to ensure a comprehensive search and screening process, several limitations must be acknowledged. First, due to a short timeline and limited non-English language proficiency of our research team, our review was restricted to articles published in English. This may have led to the omission of relevant studies published in other languages. Second, grey literature was limited to theses and dissertations. This decision reflected both time constraints and the low likelihood of retrieving additional relevant sources. Third, some researchers may consider self-report data collected within quantitative designs to constitute qualitative contributions. In the present review, however, we did not classify such studies as qualitative given their lack of paradigmatic positioning, theoretical framework, and methodology. While self-report responses may generate non-numerical data, in this context

it did not meet our operational definition of qualitative research. Finally, although our search strategy was developed in consultation with a health sciences librarian to ensure breadth and rigor, it remains possible that relevant studies were not captured by our selected search terms.

## Conclusion

This scoping review mapped and characterized the use of qualitative methods in motor imagery research. The existing literature is dominated by studies in sport psychology, with a strong emphasis on understanding athletes' use of imagery. Qualitative investigations of motor imagery in basic experimental learning contexts are notably absent. Given that laboratory research often provides the foundation for healthcare applications, the lack of qualitative insight into participants' experiences represents a critical gap. To advance the field, future work should prioritize clearer articulation of qualitative designs and deeper exploration of how individuals experience motor imagery. Such efforts are essential to ensure that applications, particularly in rehabilitation, are informed by the perspectives of those directly engaging in this inherently subjective practice. Specifically, systematic engagement with qualitative inquiry can strength the application of motor imagery in rehabilitation by centering intervention design and evaluation on patient experience. Grounding motor imagery practice in how individuals engage with and interpret imagery may enable researchers and clinicians to develop more responsive, meaningful, and ultimately more effective interventions.

## Supporting information

**S1 File. Search Strategy.** Complete search strategy, developed in collaboration with Health Sciences librarian for all databases searched.
(PDF)

**S2 File. Defining Exclusion Criteria.** Definitions of all exclusion criteria applied during the full-text screening stage.
(PDF)

**S3 File. Data Extraction Tool.** Data extraction form and accompany instructions used in the data extraction stage.
(PDF)

**S4 File. Characteristics of Studies.** Alphabetized list of all included studies with corresponded extracted data.
(XLSX)

**S5 File. PRISMA-ScR-Checklist.** Completed PRISMA-ScR checklist outlining reporting items for the scoping review.
(PDF)

## Acknowledgments

We would like to acknowledge Dalhousie University Health Sciences librarian Shelley McKibbon for her invaluable expertise and assistance in developing the search strategy for this review.

## Author contributions

**Conceptualization:** Juliet M. Rowe, Shaun G. Boe.

**Data curation:** Juliet M. Rowe, Theresa C. Gaughan.

**Formal analysis:** Juliet M. Rowe.

**Funding acquisition:** Shaun G. Boe.

**Methodology:** Juliet M. Rowe, Theresa C. Gaughan, Shaun G. Boe.

**Project administration:** Juliet M. Rowe.

**Supervision:** Shaun G. Boe.

**Visualization:** Juliet M. Rowe.

**Writing – original draft:** Juliet M. Rowe.

**Writing – review & editing:** Theresa C. Gaughan, Shaun G. Boe.

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
