## [Decision Letter · Decision Letter 0]

8 Feb 2026

PONE-D-25-67879Mapping qualitative research on motor imagery: A scoping reviewPLOS One

Dear Dr. Boe,

Thank you for submitting your manuscript to PLOS ONE. After careful consideration, we feel that it has merit but does not fully meet PLOS ONE’s publication criteria as it currently stands. Therefore, we invite you to submit a revised version of the manuscript that addresses the points raised during the review process.

We look forward to receiving your revised manuscript.

Kind regards,

Hesam Ramezanzade, Ph.D

Academic Editor

PLOS One

Journal Requirements:

2. We note that there is identifying data in the Supporting Information file S4 File.xlsx. Due to the inclusion of these potentially identifying data, we have removed this file from your file inventory. Prior to sharing human research participant data, authors should consult with an ethics committee to ensure data are shared in accordance with participant consent and all applicable local laws.

-Location data

“This work was supported by an NSERC Discovery Grant (RGPIN/04840-2020) awarded to SB and personnel support from NSERC (Canada Post Graduate Scholarship – Doctoral) and the Heart and Stroke foundation (BrightRed Student Research Award) awarded to JR. The funders were not involved in any aspects of the work or its dissemination.”

Reviewers' comments:

Reviewer's Responses to Questions

**Comments to the Author**

1. Is the manuscript technically sound, and do the data support the conclusions?

Reviewer #1: Yes

Reviewer #2: Yes

2. Has the statistical analysis been performed appropriately and rigorously? 

Reviewer #1: I Don't Know

Reviewer #2: Yes

3. Have the authors made all data underlying the findings in their manuscript fully available?

The PLOS Data policy requires authors to make all data underlying the findings described in their manuscript fully available without restriction, with rare exception (please refer to the Data Availability Statement in the manuscript PDF file). The data should be provided as part of the manuscript or its supporting information, or deposited to a public repository. For example, in addition to summary statistics, the data points behind means, medians and variance measures should be available. If there are restrictions on publicly sharing data—e.g. participant privacy or use of data from a third party—those must be specified.requires authors to make all data underlying the findings described in their manuscript fully available without restriction, with rare exception (please refer to the Data Availability Statement in the manuscript PDF file). The data should be provided as part of the manuscript or its supporting information, or deposited to a public repository. For example, in addition to summary statistics, the data points behind means, medians and variance measures should be available. If there are restrictions on publicly sharing data—e.g. participant privacy or use of data from a third party—those must be specified.requires authors to make all data underlying the findings described in their manuscript fully available without restriction, with rare exception (please refer to the Data Availability Statement in the manuscript PDF file). The data should be provided as part of the manuscript or its supporting information, or deposited to a public repository. For example, in addition to summary statistics, the data points behind means, medians and variance measures should be available. If there are restrictions on publicly sharing data—e.g. participant privacy or use of data from a third party—those must be specified.requires authors to make all data underlying the findings described in their manuscript fully available without restriction, with rare exception (please refer to the Data Availability Statement in the manuscript PDF file). The data should be provided as part of the manuscript or its supporting information, or deposited to a public repository. For example, in addition to summary statistics, the data points behind means, medians and variance measures should be available. If there are restrictions on publicly sharing data—e.g. participant privacy or use of data from a third party—those must be specified.

Reviewer #1: Yes

Reviewer #2: Yes

4. Is the manuscript presented in an intelligible fashion and written in standard English?

Reviewer #1: Yes

Reviewer #2: Yes

5. Review Comments to the Author

Reviewer #1: Re: Mapping qualitative research on motor imagery: A scoping review

General comments

The manuscript “Mapping qualitative research on motor imagery: A scoping review” aims to investigate the extent to which qualitative methods have been used in motor imagery research and to characterize how these approaches have been applied. In this manuscript, the scoping review was conducted in accordance with the JBI methodology. The authors concluded that the literature is heavily weighted toward sport psychology and understanding athletes’ use of imagery, with limited attention to how motor imagery is experienced in experimental settings. Moreover, few studies have examined the process of imagery, particularly its role in learning within laboratory contexts. There is a clear absence of qualitative insights into participants’ experiences of motor imagery, which represents an important gap in the field. Overall, this study highlights an important gap in motor imagery research.

Below are some suggestions that should be considered by the authors:

Specific comments

Introduction

Line 46: Provide references for studies that have reported conflicting results.

Line 77: You may also include the following articles:

Parimi, F., Abdoli, B., Ramezanzade, H., & Aghdaei, M. (2024). The effect of internal and external imagery on learning badminton long serve skill: The role of visual and audiovisual imagery. PLOS ONE, 19(9), e0309473.

Ramezanzade, H., Badicu, G., Cataldi, S., Parimi, F., Mohammadzadeh, S., Mohamadtaghi, M., & Greco, G. (2023). Sonification of motor imagery in the basketball jump shot: Effect on muscle activity amplitude. Applied Sciences, 13(3), 1495.

Lines 84–91: The concept of qualitative research should be explicitly introduced in the Introduction. Clarify what is meant by qualitative methods in this context.

Design

Line 115: Why was no outcome criterion included? Doesn’t the participants’ level and type of experience (e.g., children vs. adults, healthy vs. disabled) affect the quality of imagery? Please justify this decision.

Search methods

Line 130: What were the criteria for excluding articles? Please specify.

Results

Line 167: What were the inclusion criteria? Clarify explicitly.

Reviewer #2: This is solid scholarship that makes an important contribution to motor imagery research. The systematic mapping reveals genuine and consequential gaps - particularly the absence of qualitative inquiry into motor imagery experiences during learning. With revisions to deepen the critical analysis and provide more actionable guidance for future research, this paper could significantly influence how the field approaches motor imagery investigation.

1. Expand the "why" discussion: Add a section discussing possible reasons for the identified gaps (epistemological, institutional, training-related barriers).

2. Provide concrete future research recommendations: Specify what qualitative approaches would be most valuable, what questions they should address, and how they might be integrated with quantitative methods.

3. Strengthen the clinical/practical implications: Be more explicit about how these gaps matter for rehabilitation outcomes and intervention effectiveness.

4. Add methodological quality commentary: Beyond design reporting, discuss patterns in rigor, sampling, and analytical depth.

5. Discuss mixed methods: Analyze how the 8 mixed-methods studies integrated approaches and whether this model has promise.

6. Include a visual summary: Consider adding a figure showing the distribution of studies across disciplines, contexts, and research foci.

7. Make the abstract more specific about key numerical findings

8. Clarify terminology (e.g., "four Ws," "applied vs. fundamental")

9. Expand the limitations discussion

10. Improve table readability

6. PLOS authors have the option to publish the peer review history of their article (what does this mean?). If published, this will include your full peer review and any attached files.). If published, this will include your full peer review and any attached files.). If published, this will include your full peer review and any attached files.). If published, this will include your full peer review and any attached files.

...

Reviewer #1: No

Reviewer #2: No

---

## [Author Response · Author response to Decision Letter 1]

9 Mar 2026

We uploaded a file with specific reviewer and editor comments named PLOS_Response_to_Reviewers.pdf.

---

## [Editor Report · Decision Letter 1]

12 Apr 2026

Mapping qualitative research on motor imagery: A scoping review

PONE-D-25-67879R1

Dear Dr. Boe,

We’re pleased to inform you that your manuscript has been judged scientifically suitable for publication and will be formally accepted for publication once it meets all outstanding technical requirements.

Kind regards,

Hesam Ramezanzade, Ph.D

Academic Editor

PLOS One

---

## [Editor Report · Acceptance letter]

PONE-D-25-67879R1

PLOS One

Dear Dr. Boe,

I'm pleased to inform you that your manuscript has been deemed suitable for publication in PLOS One. Congratulations! Your manuscript is now being handed over to our production team.

Kind regards,

on behalf of

Dr. Hesam Ramezanzade

Academic Editor

PLOS One